# Effects of Ultrasonic Activation on Root Canal Filling Quality of Single-Cone Obturation with Calcium Silicate-Based Sealer

**DOI:** 10.3390/ma14051292

**Published:** 2021-03-08

**Authors:** Sin-Young Kim, Young-Eun Jang, Bom Sahn Kim, Eun-Kyoung Pang, Kiche Shim, Hye Ryeon Jin, Min Kyung Son, Yemi Kim

**Affiliations:** 1Department of Conservative Dentistry, Seoul St. Mary’s Hospital, College of Medicine, The Catholic University of Korea, Seoul 06591, Korea; jeui99@catholic.ac.kr; 2Department of Conservative Dentistry, College of Medicine, Ewha Womans University, Seoul 07986, Korea; jang@ewha.ac.kr (Y.-E.J.); tlarlco@hanmail.net (K.S.); jhr8141@naver.com (H.R.J.); 3Department of Nuclear Medicine, College of Medicine, Ewha Womans University, Seoul 07986, Korea; kbomsahn@ewha.ac.kr; 4Department of Periodontology, College of Medicine, Ewha Womans University, Seoul 07986, Korea; ekpang@ewha.ac.kr; 5Department of Clinical Oral Health Science, Graduate School of Clinical Dentistry, Ewha Womans University, Seoul 07986, Korea; okson23457@naver.com

**Keywords:** calcium silicate-based sealers, filling quality, micro-computed tomography, single-cone obturation, ultrasonic

## Abstract

Background: We evaluated the effects of ultrasonic activation on root canal filling quality of the single-cone (SC) obturation technique with calcium silicate sealers and gutta percha cones. Methods: Thirty-six human single-rooted premolars were obturated with gutta percha and sealer. For the continuous wave (CW) group (n = 12), AH Plus with a continuous wave technique was used. The SC group (n = 12) received EndoSequence BC sealer with a single-cone technique. The SCU (SC with the addition of ultrasonic activation) group (n = 12) received the same treatment. Micro-computed tomography was used to scan the teeth, and the void volume within the root canal was evaluated at the apical, middle, and coronal levels. Then cross-sections were observed under a light microscope and scanning electron microscope (SEM). Results: Void volume was significantly lower in the SCU group than in the CW and SC groups. There were no statistically significant differences between the CW and SC groups. The SCU group had fewer voids than the CW and SC groups in the coronal and middle third areas. Specimens showed no apparent gaps or voids in any group. SEM images revealed both gap-free and gap-containing regions at different levels in all groups. Conclusions: Single-cone obturation with calcium silicate-based sealers might obtain enhanced filling quality when used with ultrasonic activation.

## 1. Introduction

The main objective of root canal obturation is to obtain a fluid-tight seal for preventing future microbial contamination [1]. Resin-based sealers such as AH Plus have been used combined with gutta percha (GP) cones in diverse canal filling techniques. They offer advantages in solubility reduction, tight apical sealing, and sealer tag formation in the root dentin [2]. More recently, calcium silicate-based sealers have been introduced and have shown low cytotoxicity, high biocompatibility, acceptable bond strength, and sealing ability [3,4,5,6,7,8]. In addition, their potential to promote hard tissue deposition is a promising property for root canal sealers [5,6].

Based on these advances, the single-cone (SC) obturation technique using calcium silicate sealers has been suggested as an alternative method to the continuous wave (CW) technique using epoxy resin-based sealers [9,10]. The SC technique with calcium silicate sealers is considered less sensitive and has shown equivalent or superior sealing ability compared to the CW technique [9,11]. However, the SC technique without apical pressure results in more voids in the cervical third area than the CW technique [12]. There have been conflicting results regarding the quality of root canal filling based on various obturation techniques and sealer materials [9,12,13]. Compared to the CW technique, the SC technique generally allows for more sealer and less GP. It is advantageous to obturate teeth with irregular root canals; however, porosity may occur when large volumes of sealers are applied. Proper application of GP cones and sealer on dentin surface remains challenging. The use of ultrasonic energy may improve the quality of root canal fillings using calcium silicate cement and calcium silicate-based sealers [13,14]. However, the effectiveness of ultrasonic activation is still controversial [15]. Moreover, few studies have investigated the incidence of voids in various canal filling techniques utilizing ultrasonic application.

Therefore, we evaluated the effects of indirect ultrasonic activation on root canal filling quality using the SC obturation technique with calcium silicate sealers and gutta percha cones. The null hypothesis was that there would be no significant differences in the proportions of voids in root canals between the various obturation techniques.

## 2. Materials and Methods

### 2.1. Preparation of Tooth Samples

Thirty-six human single-rooted premolars were obtained with the informed consent of patients, and all protocols were approved by the Institutional Review Board (IRB) of the Ewha Womans University Hospital (EUMC 2018-03-036). The following teeth were included in this study: teeth with no previous root canal treatment, and no signs of cracks, perforations, internal or external resorption, or root caries. Pieces of crown were removed using a diamond saw (Minitom; Struers, Copenhagen, Denmark) to form standardized root samples of 11 mm length. Teeth were examined under a microscope (Carl Zeiss Meditec AG, Oberkochen, Germany) at 20× magnification to exclude cracks, perforations, internal resorptions, external resorptions, and root caries. Teeth with oval-shaped canals only were used in this study.

A size #10 K-file (Dentsply Maillefer, Ballaigues, Switzerland) was used to gain canal patency by inserting into each root canal until the tips were visible at the apical foramen. The working length was established by reducing 0.5 mm from the length measured. The root canals were instrumented to an apical size #35/0.06 taper with a crown-down technique using #15 K-file (Dentsply Maillefer) and ProFile (Dentsply Maillefer) Ni-Ti rotary instruments in the presence of a 2.5% sodium hypochlorite (NaOCl) solution. Then, the canals were immersed in 10 mL 17% EDTA for 2 min, followed by 3 mL 5.25% NaOCl for 5 min. An irrigation with 10 mL distilled water using Endo-Eze^®^ 27-gauge side-cut irrigation needles (Ultradent Products, Inc., South Jordan, UT, USA) was performed, and the canal was dried with four sterile paper points (Dentsply Maillefer) for 3 s.

The teeth were randomly distributed into the following three groups (n = 12) before obturating the instrumented canal spaces: the CW group (continuous wave technique using AH Plus), SC group (single-cone technique using EndoSequence BC sealer), and SCU group (same as SC but with the addition of ultrasonic activation).

In the CW group, each canal was obturated with a GP cone (Diadent Group International, Cheongju, Korea) and AH Plus (Dentsply DeTrey, Konstanz, Germany) by using the continuous wave compaction technique. Briefly, a GP cone and sealer were placed into the root canal. Then, down-packing with a System B (SybronEndo, Orange, CA) was performed to within 3 mm of the working length, and the canal was backfilled using SuperEndo-Beta (B&L BioTech, Alexandria, VA, USA).

In the SC group, EndoSequence BC sealer (Brasseler USA, Savannah, Georgia, USA) was applied directly into the canal using an intra-canal tip supplied by the manufacturer. The GP cone was gently inserted into the canal with an up-and-down motion until it reached the working length, and excess GP was cut at the canal orifice level using a System B plugger (SybronEndo, Orange, CA, USA).

In the SCU group, after applying the sealer and GP cone into the canal, a smooth ultrasonic tip (StartX #3, Dentsply Maillefer) was placed into contact with the cotton plier holing the GP cone. Then, it was activated for 2–3 s on the lowest power setting on the piezoelectric ultrasonic unit (Obtura-Spartan, Fenton, MO) until the GP cone reached the working length. Excess GP was cut at the level of the canal orifice using a System B plugger (SybronEndo, Orange, CA, USA).

Finally, a bonding agent (Clearfil SE; Kuraray Dental, Tokyo, Japan) was applied with agitation for 20 s and light cured (Elipar™ DeepCure-S LED Curing Light; 3M Dental Products, St.Paul, MN, USA) for 20 s. The access cavities were filled with a flowable composite resin (Tetric N-flow; Ivoclar Vivadent, Schaan, Liechtenstein), and the teeth were stored at 37° C with 100% humidity (SANYO Incubator MCO-175, Osaka, Japan) for 14 days for the complete setting of the sealers.

### 2.2. Micro-CT Evaluation

All teeth in three groups (n = 12) were scanned by a high-resolution micro-CT system (SkyScan 1173 X-ray scanner, Bruker-Micro-CT, Kontich, Belgium), and each root was scanned from the anatomic apex to 11 mm superior with the same setting values: an aluminum filter 0.5 mm thick, a spatial resolution of 9.94 µm at 110 kV and 72 mA, a 360° rotational angle, a 0.3° rotation step, and a 500 ms exposure time. Reconstruction was performed under a 10% beam hardening correction and a ring artifact reduction value of 10. Scanned images were reconstructed using the Data Viewer 64 software (version 1.5.2.4: Bruker), and CT-An (version 1.16.1; Bruker) was used to analyze void volumes.

The voids were assessed in two-dimensional (2D) slices. Cross-sectional images were acquired in the coronal, axial, and sagittal planes. Axial sections taken perpendicular to the longitudinal axis of the root were evaluated by a single blinded (to canal filling methods) observer. The gray scale images were converted into binary images by image thresholding to gaps and voids from the sealers, gutta percha, and root dentin. The grayscale ranges of each object to be recognizable were determined using a density histogram with a global threshold method [12]. The original and segmented images were thoroughly examined to confirm segmentation accuracy. The area of gaps and voids from the apex (designated as 0 mm-level) to the 11 mm-level of each axial slice of tooth was evaluated.

A three-dimensional (3D) analysis of the volume of gaps and voids was conducted within the volume of interest (VOI). The volumetric percent distributions of gaps and voids in the three groups were analyzed for the VOI (0-11 mm), and subgroup analyses were performed at the apical (0–3 mm), middle (3–7 mm), and coronal (7–11 mm) levels [16]. All measurements were repeated after an interval of 1 week, and the average of the data was used for analyses.

The 3D percent volume of the voids was analyzed using one-way analysis of variance (ANOVA) to examine the differences among the obturation techniques. Tukey’s multiple comparison tests were also performed. All statistical analyses were performed using SPSS software (ver. 21; SPSS, Inc., Chicago, IL, USA) with a significance level set at *p* < 0.05.

### 2.3. Light Microscopy and Scanning Electron Microscopy

After micro-CT scanning, a tooth from each group was randomly selected and sectioned with increments of 1 mm starting from the apex using a slow-speed diamond saw under water cooling (Minitom; Struers, Denmark). The cross-sections were further examined using a light microscope at a minimum of 20× magnification.

Two teeth from each group were randomly selected and sectioned horizontally and longitudinally for SEM analyses. The specimens were vacuum dried, coated with gold by ion sputter (IB-3, Eiko, Japan), and observed by scanning electron microscopy (SEM) (Sigma 300; Zeiss, Oberkochen, Germany) at magnifications of 200, 1000, and 2000×.

## 3. Results

The reconstructive images of the root canal fillings are shown in Figure 1A,B). The representative 2D percent area distributions of voids in axial sections along the 0–11 mm range from the apex are shown in Figure 1C. The SCU group had the lowest void area distribution.

Figure 1D shows the results of 3D quantification of void volume. The void volumes were significantly lower in the SCU group than in the CW and SC groups. There were no statistically significant differences between the CW and SC groups. In subgroup analyses, the SCU group had fewer voids than the other groups in the cervical and middle third areas. In the apical third region, the SCU group had lower void volumes than those of the CW group.

In light microscopy images, sealers were observed to be in close contact with the canal walls, and there were no apparent gaps or voids in the cross-sections of any group (Figure 2). In the SEM images, there were both gap-free and gap-containing regions at different levels in all groups (Figure 2 and Figure 3).

## 4. Discussion

Achieving a 3D hermetic seal is the main goal of root canal filling. Root canal obturation without voids correlates with a high success rate of root canal treatment [17]. Thus, minimal gap formation is the most important feature of root canal sealers. In assessing gaps, leakage tests have several limitations, such as non-reproducibility, large standard deviations, and high variability in results [18]. Micro-CT imaging for evaluating gaps could be an alternative method because it provides qualitative and quantitative results and is nondestructive [19,20].

The SC obturation technique with calcium silicate sealers and gutta percha cones can be differentiated from a traditional SC technique. This approach seems promising because calcium silicate sealers show good sealing ability and biocompatibility with hydrophilic properties [6,8,21]. However, there are some concerns regarding the presence of gaps and voids, which can negatively affect a successful endodontic outcome [22,23,24]. In addition, there have been a limited number of studies with respect to application methods of SC obturation. Therefore, we evaluated the effects of ultrasonic activation on root canal filling quality using the SC obturation technique with calcium silicate sealers and gutta percha cones. Micro-CT analyses, light microscopy, and SEM were used to evaluate the formation of gaps and voids.

The micro-CT results showed reduced void volumes for the SC technique when accompanied by ultrasonic activation. This result is in contrast to a previous study that showed no significant difference in void volumes between SC and SCU groups. However, in that study, SCU groups showed a lower number of voids and scores compared to SC-only groups based on the stereomicroscopic examination of sectioned samples [13]. More research is needed, as the methodology and the characteristics of tested sealer materials can affect the quality of root canal filling. When using ultrasonication, gentle vibration is recommended, because a previous study reported that excessive ultrasonic energy may accelerate void formation and negatively affect filling quality [14]. To date, there are no guidelines on application methods of ultrasonic energy when it is used for the SC technique. Further studies on appropriate application methods are needed to clarify if ultrasonic activation affects the filling quality and possible extrusion of sealer.

In comparing the void volumes between the SC and CW groups, in accordance with previous studies, our results show no significant difference in void volumes between groups [9,25,26]. However, some studies have reported increased void volumes using the SC technique compared to the CW technique, particularly in the cervical third area [12]. These conflicting results may be due to the limitation of micro-CT; the radiopacity of sealers may affect the results [13].

Interestingly, the CW group had more voids than the SC groups at the 1 mm level from the apex in 2D analyses (Figure 1C). These results are consistent with previous studies that have shown that the penetration depth of System B pluggers affects canal filling quality using the CW technique [27,28]. In such studies, >90% of gutta percha-filled area was observed only at the 1 mm level distal from the plugger tip. It is not easy for the pluggers to reach 1 mm from the apex because of practical issues, such as a possible extrusion of gutta percha and the limit of the appropriate width of System B pluggers. Moreover, application of excessive compaction forces might cause root fractures, particularly in teeth suspected of having a crack [27,28].

The microscopic examination of cross-sectioned samples showed that there were no apparent gaps or voids, which is in contrast to a previous study [13]; however, the SEM images of the present study showed gaps in some sections. These contradictory results can be explained by the limitations of the sectioning method, such as the loss of materials during cutting, sample selection bias, and the different magnifications and resolution of the two approaches. Gaps and voids can adversely affect the success rate of root canal treatment, as porosity in the filled root canals can harbor contaminants and lead to the regrowth of microorganisms [29]. The results of the present study demonstrate that the indirect application of ultrasonic energy can reduce the volume of gaps and voids when using the SC technique, which is associated with favorable endodontic outcomes. However, long-term clinical studies regarding the clinical success rate of this technique are needed.

## 5. Conclusions

The SC technique with ultrasonic activation led to fewer voids than in the CW- and SC-only techniques in the cervical and middle third areas. Within the limitations of this study, SC obturation with calcium silicate-based sealers resulted in enhanced filling quality when used with ultrasonic activation.

## Figures and Tables

**Figure 1 materials-14-01292-f001:**
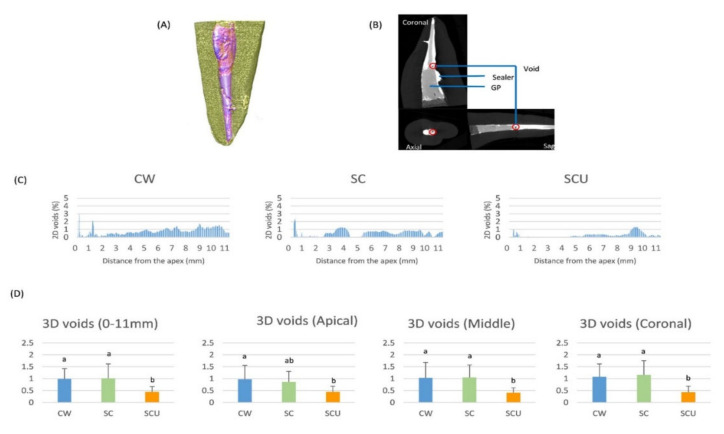
Micro-CT images of a representative specimen and Micro-CT analyses of 2D and 3D quantifications of void volumes. (**A**) Representative 3D images produced from the grayscale thresholds of filling materials. (**B**) Representative 2D images of coronal, axial, and sagittal views, which are shown for the volume of interest (VOI). The root structure, gutta percha (GP), and sealer can be differentiated by the value of their different grayscale levels. Interfacial gaps and voids can be identified within the root filling. (**C**) Examples of 2D area of gaps and voids in the roots obturated by AH Plus with the continuous wave technique (CW group), EndoSequence BC sealer with the single-cone technique (SC group), and EndoSequence BC sealer with the single-cone technique accompanied by ultrasonic activation (SCU group). (**D**) The 3D percent volume of voids within the entire root, from the apex (0 mm) to 11 mm coronal to the apex, apical (0–3 mm) level, middle (3–7 mm) level, and coronal (7–11 mm) level. Bars identified with the same lowercase letter are not significantly different (*p* > 0.05).

**Figure 2 materials-14-01292-f002:**
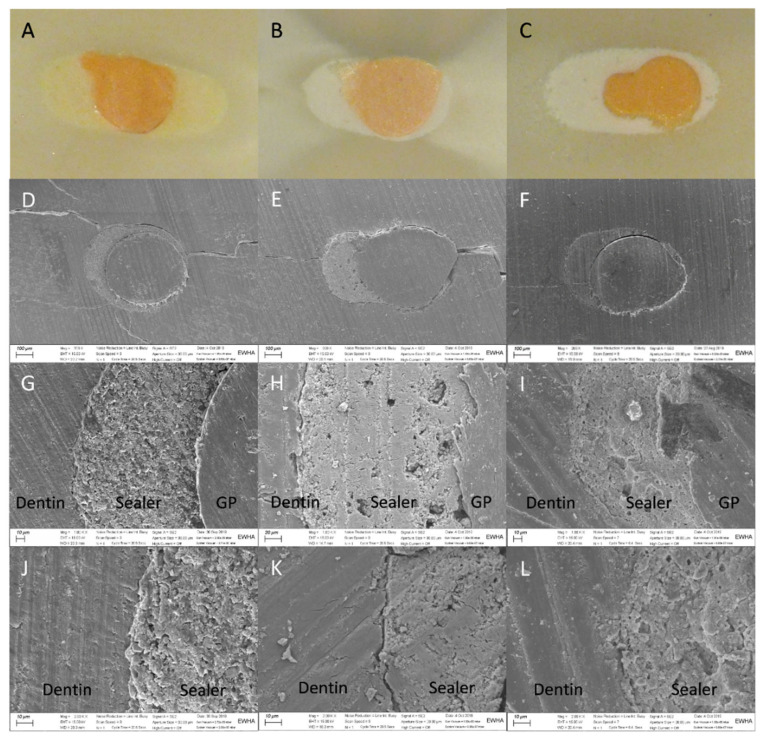
Light microscopy and SEM images of the dentin–sealer interface taken at magnifications of 20, 200, 1000, and 2000×. (**A**,**D**,**G**,**J**) AH Plus with the continuous wave technique (CW group), (**B**,**E**,**H**,**K**) EndoSequence BC sealer with the single-cone technique (SC group), and (**C**,**F**,**I**,**L**) EndoSequence BC ultrasonic with the single-cone technique accompanied by ultrasonic activation (SCU group).

**Figure 3 materials-14-01292-f003:**
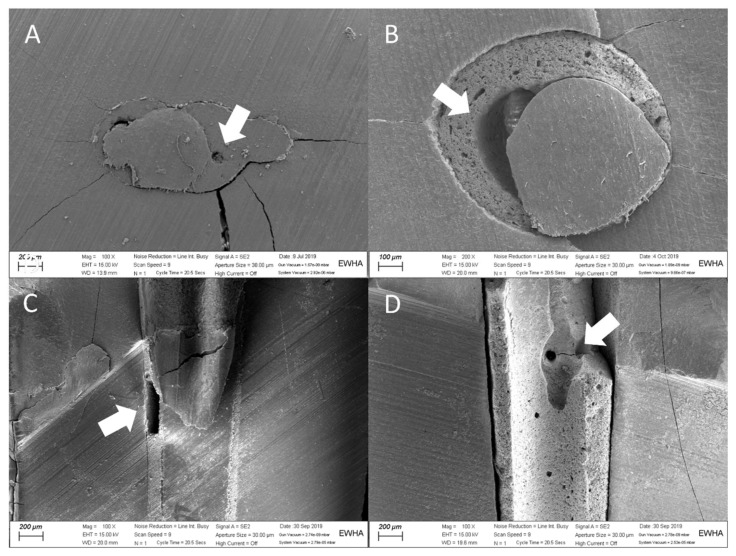
SEM images showing interfacial gaps and voids (arrows): (**A**) 100× horizontal section of EndoSequence BC sealer with the single-cone technique (SC group), (**B**) 200× horizontal section of EndoSequence BC sealer with the single-cone technique (SC group), (**C**) 100× sagittal section of AH Plus with the continuous-wave technique (CW group), and (**D**) 100× sagittal section of EndoSequence BC sealer with the single-cone technique (SC group).

## Data Availability

Not applicable.

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
