# Peer review of "Effects of Ultrasonic Activation on Root Canal Filling Quality of Single-Cone Obturation with Calcium Silicate-Based Sealer"

_materials, 2021, doi:10.3390/ma14051292_

Round 1

Reviewer 1 Report

The article is well written and the topic if of interest.

Is it possible to quantify "gentle vibration" you specify in the discussion with a value to give the reader a range for clinical application? And also what "excessive energy" may stand for?

I would suggest to add the following reference to your manuscript making some reference with it (some materials are the same):

De Bem IA, de Oliveira RA, Weissheimer T, Bier CAS, Só MVR, Rosa RAD. Effect of Ultrasonic Activation of Endodontic Sealers on Intratubular Penetration and Bond Strength to Root Dentin. J Endod. 2020 Sep;46(9):1302-1308. doi: 10.1016/j.joen.2020.06.014. Epub 2020 Jun 29. PMID: 32615175.

Author Response

Thank you for giving me the opportunity to submit a revised draft of my manuscript. We are grateful to editor and reviewers for their insightful comments on our paper. We have been able to incorporate changes to reflect most of the suggestions provided by the reviewers. We have highlighted the changes within the manuscript.

Here is a point-by-point response to the reviewer’s comments and concerns.

Point 1: The article is well written and the topic if of interest. Is it possible to quantify "gentle vibration" you specify in the discussion with a value to give the reader a range for clinical application? And also what "excessive energy" may stand for? I would suggest to add the following reference to your manuscript making some reference with it (some materials are the same):

De Bem IA, de Oliveira RA, Weissheimer T, Bier CAS, Só MVR, Rosa RAD. Effect of Ultrasonic Activation of Endodontic Sealers on Intratubular Penetration and Bond Strength to Root Dentin. J Endod. 2020 Sep;46(9):1302-1308. doi: 10.1016/j.joen.2020.06.014. Epub 2020 Jun 29. PMID: 32615175.

Response 1: Thank you for your suggestion. The term of “gentle vibration” was specified to “gentle vibration with lowest power setting.” In addition, “excessive energy” was changed into “high power ultrasound.” Please see line 230.

Point 2: I would suggest to add the following reference to your manuscript making some reference with it (some materials are the same):

De Bem IA, de Oliveira RA, Weissheimer T, Bier CAS, Só MVR, Rosa RAD. Effect of Ultrasonic Activation of Endodontic Sealers on Intratubular Penetration and Bond Strength to Root Dentin. J Endod. 2020 Sep;46(9):1302-1308. doi: 10.1016/j.joen.2020.06.014. Epub 2020 Jun 29. PMID: 32615175.

Response 2: Thank you for your comment. The reference was added to the manuscript according to your suggestion. Please see line 59 and 323.

Reviewer 2 Report

This article has evaluated the effects of ultrasonic activation on root canal filling quality of the single-cone (SC) obturation technique with calcium silicate sealers and gutta percha cones. The authors have compared three different kinds of obturation techniques, and divided into the CW group (continuous wave technique using AH plus), SC group (single-cone technique), and SCU group (same as SC but with the addition of ultrasonic activation), respectively. Through micro-CT evaluation and light microscopy and scanning electron microscopy, The SC technique with ultrasonic activation led to few voids than the CW and SC in the cervical and middle third areas. As the effectiveness of ultrasonic activation is still controversial, the article has showed SC obturation with calcium silicate-based sealers resulted in enhanced filling quality when used with ultrasonic activation with proper methods and analysis. This article is coherent and fluent and with a clear structure.

However, there is a lack of illustration of statistical analysis in Materials and Method. As proper analysis of statistic is important to discuss the result, how to analysis the statistic should be added. And I think it would better to show the difference in void volume in reconstructed 3D-images, which is more visible and distinct. I think this article can be accepted after minor revision.

Author Response

Thank you for giving me the opportunity to submit a revised draft of my manuscript. We are grateful to editor and reviewers for their insightful comments on our paper. We have been able to incorporate changes to reflect most of the suggestions provided by the reviewers. We have highlighted the changes within the manuscript.

Here is a point-by-point response to the reviewer’s comments and concerns.

Point 1: This article has evaluated the effects of ultrasonic activation on root canal filling quality of the single-cone (SC) obturation technique with calcium silicate sealers and gutta percha cones. The authors have compared three different kinds of obturation techniques, and divided into the CW group (continuous wave technique using AH plus), SC group (single-cone technique), and SCU group (same as SC but with the addition of ultrasonic activation), respectively. Through micro-CT evaluation and light microscopy and scanning electron microscopy, The SC technique with ultrasonic activation led to few voids than the CW and SC in the cervical and middle third areas. As the effectiveness of ultrasonic activation is still controversial, the article has showed SC obturation with calcium silicate-based sealers resulted in enhanced filling quality when used with ultrasonic activation with proper methods and analysis. This article is coherent and fluent and with a clear structure.

However, there is a lack of illustration of statistical analysis in Materials and Method. As proper analysis of statistic is important to discuss the result, how to analysis the statistic should be added.

Response 1: Thank you for your suggestion. We added the separate section for statistical analyses. Please see line 157.

Point 2: And I think it would better to show the difference in void volume in reconstructed 3D-images, which is more visible and distinct. I think this article can be accepted after minor revision.

Response 2: Thank you for your comment. The reconstructed 3D-images are as follows. We did not show that in the manuscript since the quantification of total void volume (fig.1, see in attach) could present the differences more clearly.

Reviewer 3 Report

A job well done and with some very nice photographs.

IN line 83: "with a crown-deown", must be changed to "crown down"

In line: 85 : "canals were immersed with 10 mL 17% EDTA for 2 min" , is not well explained, I guess they should mean "irrigated", I suggest you change it

The temperature and time parameters for the sound pack and backfill should be well specified in the material and methods.

They would have to specify in the article how the gutta-percha cones were calibrated

In the distribution of the groups, the initial volumes of the groups of each sample should be indicated in order to establish a correct comparison.

The authors should justify why they have not used EndoSequence BC Sealer HiFlow cement instead of conventional BC sealer, as ek HiFlow is designed to be used with heat-based thermoplastic techniques. Several studies have shown that conventional BC sealer cement should not be used with continuous wave.

Author Response

Thank you for giving me the opportunity to submit a revised draft of my manuscript. We are grateful to editor and reviewers for their insightful comments on our paper. We have been able to incorporate changes to reflect most of the suggestions provided by the reviewers. We have highlighted the changes within the manuscript.

Here is a point-by-point response to the reviewer’s comments and concerns.

Point 1: A job well done and with some very nice photographs.

IN line 83: "with a crown-deown", must be changed to "crown down"

Response 1: Thank you for your comment. We corrected the typo.

Point 2: In line: 85 : "canals were immersed with 10 mL 17% EDTA for 2 min" , is not well explained, I guess they should mean "irrigated", I suggest you change it.

Response 2: Thank you for your suggestion. That part was changed according to your advice.

Point 3: The temperature and time parameters for the sound pack and backfill should be well specified in the material and methods.

Response 3: Thank you for your suggestion. The temperature and time parameters were stated in the material and methods section. Please see line 98.

Point 4: They would have to specify in the article how the gutta-percha cones were calibrated

Response 4: Thank you for your comment. We used gutta-percha measurement gauge and stated that in the manuscript. Please see line 95.

Point 5: In the distribution of the groups, the initial volumes of the groups of each sample should be indicated in order to establish a correct comparison.

Response 5: Thank you for your suggestion. The base line volumes of each goups were indicated in the manuscript. Please see line 94.

Point 6: The authors should justify why they have not used EndoSequence BC Sealer HiFlow cement instead of conventional BC sealer, as ek HiFlow is designed to be used with heat-based thermoplastic techniques. Several studies have shown that conventional BC sealer cement should not be used with continuous wave.

Response 5: Thank you for your comment. We used AH Plus for CW technique since it has been regarded as a gold standard for several years. Comparison between Endosequence BC HiFlow and AH Plus would be a topic of interest, and further study regarding that is needed.

Round 2

Reviewer 3 Report

Thank you for the correction

It would be advisable to justify correctly in the discussion the use of the BC sealer instead of the BC Hiflow. Or use previous articles published using bc sealer.